# Mapping the ultrafast flow of harvested solar energy in living photosynthetic cells

Peter D. Dahlberg[1], Po-Chieh Ting[2], Sara C. Massey [2], Marco A. Allodi[2], Elizabeth C. Martin[3], C. Neil Hunter [3] & Gregory S. Engel [2]

Photosynthesis transfers energy efficiently through a series of antenna complexes to the reaction center where charge separation occurs. Energy transfer in vivo is primarily monitored by measuring fluorescence signals from the small fraction of excitations that fail to result in charge separation. Here, we use two-dimensional electronic spectroscopy to follow the entire energy transfer process in a thriving culture of the purple bacteria, *Rhodobacter sphaeroides*. By removing contributions from scattered light, we extract the dynamics of energy transfer through the dense network of antenna complexes and into the reaction center. Simulations demonstrate that these dynamics constrain the membrane organization into small pools of core antenna complexes that rapidly trap energy absorbed by surrounding peripheral antenna complexes. The rapid trapping and limited back transfer of these excitations lead to transfer efficiencies of 83% and a small functional light-harvesting unit.

[1] Graduate Program in the Biophysical Sciences, Institute for Biophysical Dynamics, and the James Franck Institute, The University of Chicago, Chicago, IL 60637, USA. [2] Department of Chemistry, Institute for Biophysical Dynamics, and the James Franck Institute, The University of Chicago, Chicago, IL 60637, USA. [3] Department of Molecular Biology and Biotechnology, University of Sheffield, Firth Court, Western Bank, Sheffield, S10 2TN, UK. Correspondence and requests for materials should be addressed to G.S.E. (email: gsengel@uchicago.edu)

Photosynthesis relies on ultrafast energy transfer to efficiently move energy from the site of absorption (photosynthetic antenna) to the site of charge separation (photosynthetic reaction center, RC) on a picosecond timescale[1, 2]. These processes are crucial for the organism's fitness and are tightly regulated and optimized for safe harvesting of solar energy[3–6]. For the better part of a century, in vivo energy transfer has been monitored by measuring the small percentage of absorbed light emitted as fluorescence. Here, we reveal the fate of the majority of absorbed excitations in a living bacterial cell, using two-dimensional electronic spectroscopy (2DES) to follow the flow of energy in the model organism Rhodobacter sphaeroides.

The light-harvesting machinery in Rba. sphaeroides is composed of two different types of antenna complexes (Fig. 1a)[7, 8]. Light-harvesting complex 2, LH2, has two rings of bacteriochlorophyll a (Bchl a). One ring contains 9 weakly coupled Bchl a that absorb at around 800 nm (Fig. 1b), known as the B800 band, and the other ring contains 18 strongly coupled Bchl a that absorb at around 850 nm, known as the B850 band[7]. Light-harvesting complex 1, LH1, dimerizes to form an "S" shaped array of 56 Bchl a that encircles two RCs. Dimerization is driven by the protein PufX that links two LH1s together at a slight angle, driving membrane curvature[8]. The 56 Bchl a in an LH1 dimer are strongly coupled and form a complex electronic structure with a

dominant absorption feature at 875 nm, known as the B875 band. Studies on membrane fragments suggest that LH2 transfers energy to LH1 on a few picoseconds timescale and LH1, in turn, transfers energy to the special pair of Bchl a in the RC, the site of charge separation[9–13]. The transfer between LH1 and the special pair is energetically uphill, from 875 to 870 nm. This transfer is the rate-limiting step in the energy transfer process, occurring within approximately 50 picoseconds[11, 14].

In this work, we overcome the longstanding obstacle of intensely scattered light[15, 16] and recover the native dynamics of energy transfer through the dense network of antenna complexes and into the RC. 2DES spectra correlate excitation energy along the $\omega_\tau$-axis with stimulated emission (SE), ground state bleach (GSB), and excited state absorption (ESA) energies along the $\omega_t$-axis as a function of an ultrafast time delay, T, known as the waiting time (Supplementary Fig. 1)[17, 18]. Here, we use the GRadient Assisted Photon Echo Spectrometer (GRAPES), which multiplexes one of the sampling dimensions[19]. Multiplexing reduces signal acquisition time by several orders of magnitude and allows fine sampling of the waiting time domain. This fine sampling of the waiting time enables the removal of scattered light in Fourier domains inaccessible by conventional 2DES[16]. The energy transfer dynamics observed in vivo in conjunction with simulations constrain the membrane organization into small

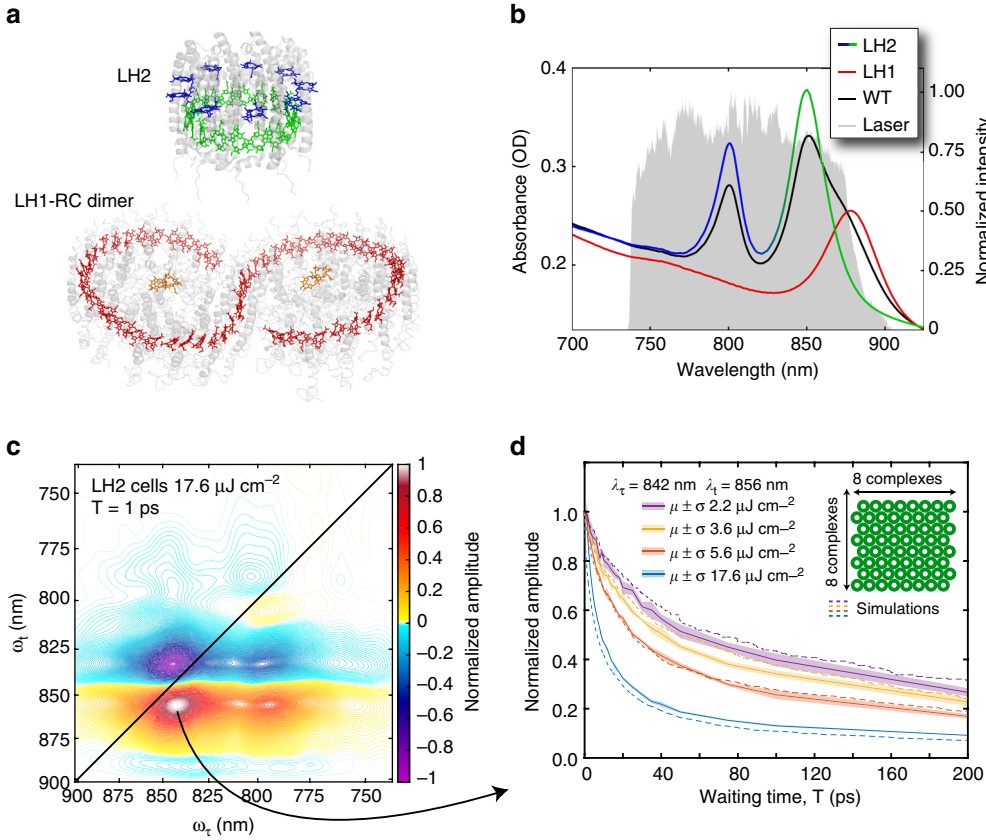

**Fig. 1** Annihilation reveals a highly connected network of light-harvesting complexes in vivo. **a**, Crystal structures of LH2 and RC-LH1-PufX (PDB 1KZU and PDB 4JC9, respectively) with the carotenoids and Bchl a phytyl tails removed for clarity. LH2 contains two bands of Bchl a, the B800 (blue) and the B850 (green). LH1 contains a single band of Bchl a, B875 (red) that transfers energy to the special pair of the RC (orange). **b**, Absorption spectra in a 200 μm path length of cells containing only LH2, cells containing only LH1, and wild-type (WT) cells. The large offset from zero optical density is due to optical scattering. The 2DES excitation spectrum is shown in gray and is produced by super continuum generation in argon gas. The spectrum is broad enough to interrogate the entire energy transfer process from LH2→LH1→RC. **c**, Absorptive 2DES spectrum of LH2-only cells taken with 17.6 μJ cm$^{-2}$ at T = 1 ps. **d**, Waiting time traces acquired at different powers from the maximum of the GSB/SE feature. The traces are the average of three scans and the shaded background is the mean ± the standard deviation. The change in dynamics with power is indicative of exciton-exciton annihilation. The dashed traces are the population of excited LH2 from a random walk simulation with a lifetime for energy transfer between LH2s of 2.7 ps, a domain size of 64 LH2, and a fluorescence lifetime of 250 ps

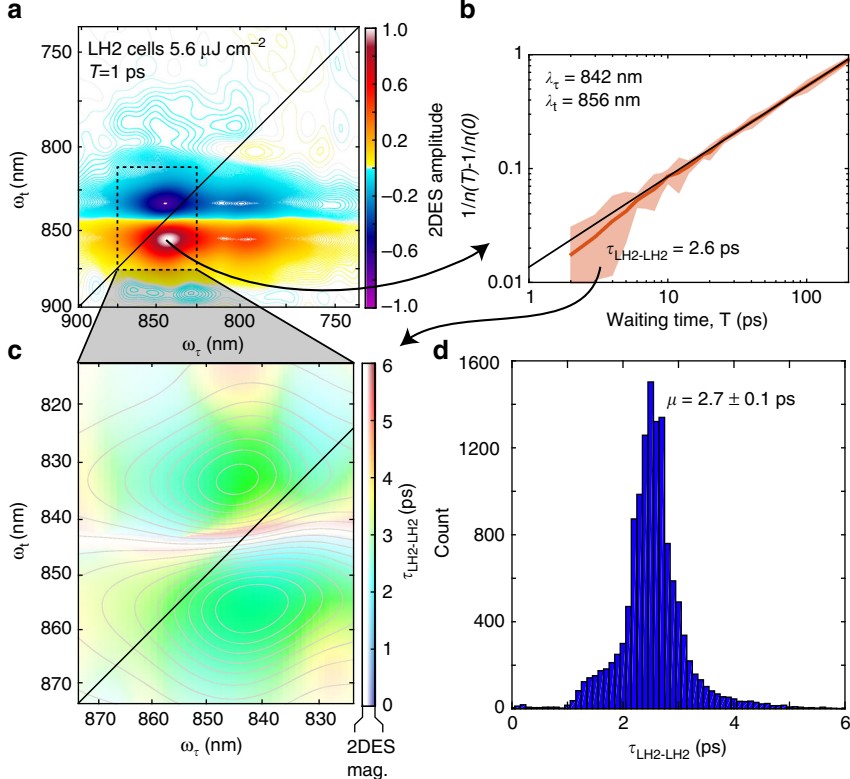

**Fig. 2** Annihilation dynamics constrain energy transfer times between isoenergetic antenna. **a** Absorptive 2DES spectrum of LH2-only cells at T = 1 ps collected at 5.6 μJ cm⁻². The *dashed* box is analyzed further for the lifetime of energy transfer between LH2 complexes. **b** Waiting time dynamics from the maximum GSB/SE feature presented following equation 2, where the 2DES intensity is $n(T)$. The intercept of the linear relationship is used to retrieve the annihilation rate, $\gamma_0$, which is used to recover the hopping time, $\tau_{hop}$, given in equation 1. **c** Color map of the recovered $\tau_{hop}$. The contours and saturation of the color are given by the intensity of the 2DES signal at 1 ps. **d** Histogram of the recovered lifetimes giving a mean of 2.7 ps for the lifetime of energy transfer between LH2s

pools of core antenna complexes that rapidly trap energy absorbed by surrounding peripheral antenna complexes. These results agree with previous atomic force microscopy (AFM)[20] and ultrafast spectroscopy studies on isolated complexes and membrane fragments[9–14, 21], as well as simulations and theoretical studies[22–25]. Additionally, these results answer longstanding questions concerning the energy transfer dynamics and membrane organization truly present in living cells and are an essential demonstration that the isolation processes employed for decades do not significantly perturb the energy transfer process. Further, our measurements establish the powerful capabilities of this variant of 2DES. These capabilities can be broadly applied to more complex photosynthetic organisms such as plants and algae where numerous questions remain surrounding membrane architecture, dynamics, and regulatory processes.

## Results
**Annihilation dynamics in mutants with only LH1 or LH2.** Measuring energy transfer dynamics in well-connected systems of identical subunits, like those found in photosynthetic membranes, presents two key experimental challenges. First, much of the energy transfer occurs between isoenergetic complexes that are spectrally indistinguishable from one another. Second, even at relatively low excitation concentrations, exciton-exciton annihilation can dominate observed dynamics[2, 26, 27]. Performing a power-dependence study of dynamics with excitation fluences of 17.6, 5.6, 3.6, and 2.2 μJ cm⁻² per pulse allows us to identify annihilation processes by observing changes in dynamics between excitation fluences; we can then exploit the annihilation dynamics

to determine energy transfer times between the isoenergetic complexes. Figure 1c shows the absorptive 2DES spectrum of cells containing only LH2 complexes[28] at our highest excitation fluence of 17.6 μJ cm⁻². The waiting time traces shown in Fig. 1d show clear power-dependent dynamics. From the molar extinction coefficient of LH2 and the excitation spectrum, we calculate that the lowest excitation fluences of 3.6 and 2.2 μJ cm⁻² correspond to excitation of ~1 in every 30 and ~50 LH2 complexes, respectively. The changing dynamics between these fluences is indicative of exciton-exciton annihilation and suggests a large number of connected LH2 complexes, in agreement with previous studies[27]. The same power-dependent study is reproduced for cells containing only LH1 complexes[28] in Supplementary Fig. 2 and shows annihilation only at the highest fluence corresponding to excitation of ~1 in every 8 LH1 complexes.

The annihilation dynamics are used to recover energy transfer times between isoenergetic complexes, which are normally unobtainable in 2DES studies because they yield no cross peaks or dynamic signals. Following the analysis of Barzda et al.[2, 26] on aggregates of an isolated plant antenna, the annihilation rate, $\gamma_0$, is related to the transfer time, $\tau_{hop}$, by

$$\gamma_0^{-1} = 0.5 N f_d(N) \tau_{hop} \qquad (1)$$

where $N$ is the number of connected antenna complexes, in this case LH2, and $f_d(N)$ is the structure factor indicative of the packing arrangement. At large values of $N$, $f_d(N)$ depends weakly on $N$ and is determined by the assumed arrangement of complexes[26]. Based on previous AFM studies[20], we assume an arrangement between a square lattice, $f_d = 0.8$, and a hexagonal

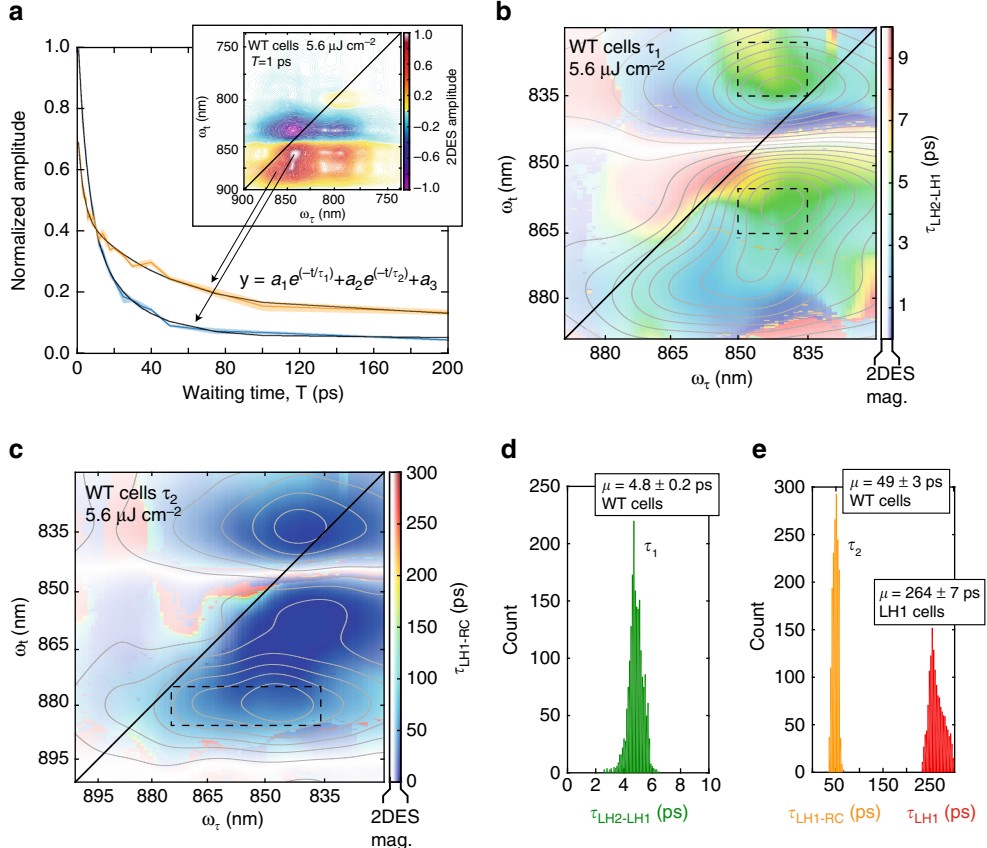

**Fig. 3** Energy transfer and trapping in the complex network of light-harvesting machinery. **a** Waiting time traces taken from the peak locations of GSB/SE features of LH2 (*blue*) and LH1(*orange*) in the wild-type *Rba. sphaeroides*. The 2DES spectra were normalized to 1 ps and fit to a biexponential function. The shaded background is the mean ± the standard deviation. **b** Color map of the first lifetime from the biexponential fit. This lifetime in the region of LH2 corresponds to the energy transfer from LH2 to LH1. The saturation of the color as well as the gray contours is given by the intensity of the 2DES spectrum at T = 1 ps. **c** same as **b** except for the second lifetime in the biexponential fit. The saturation and contours are coming from the T = 50 ps spectrum and the lifetime corresponds to LH1-RC energy transfer. **d** Histogram of the lifetimes within the dashed boxes of **b** These lifetimes correspond to LH2→ LH1 transfer times. **e** Histogram of the lifetimes within the dashed box of **c**. These lifetimes correspond to LH1→ RC transfer. The *red histogram* is from LH1-only cells and represents the fluorescence lifetime of LH1, 264 ± 7 ps SEM, and is consistent with the lifetime recovered from modeling

lattice, $f_d = 0.6$, and take $f_d(N)$ to be a constant value of 0.7. The 2DES signal can then be related to the annihilation rate via

$$\log\left(\frac{1}{n(T)} - \frac{1}{n(0)}\right) = (d_s/2)\log(T) + \log\left(\frac{\gamma_0}{d_s}\right) \quad (2)$$

where $n(T)$ is the population of excitations at waiting time $T$ and $d_s$ is the fractal dimension, typically around 1.8, which accounts for restriction of diffusion in the membrane to the lattice of antenna complexes[29].

Figure 2 illustrates the experimental procedure for recovering the energy transfer time between LH2s. Waiting time traces from the data collected at a fluence of 5.6 μJ cm$^{-2}$ are fit to the linear relationship of equation 2 and are shown in Fig. 2b. These fits are performed for each point within the dashed box in Fig. 2a. The recovered lifetime of 2.7 ± 0.1 ps, where ± denotes the standard error of the mean (SEM), is homogenous and the same for the ESA feature above the diagonal and the GSB/SE feature below the diagonal. Data collected at any of these excitation fluences could have been used to estimate the energy transfer time between LH2s; however, using the highest fluence risks initially populating LH2s with multiple excitations that would give rise to annihilation not due to inter-complex transfer, while the lowest fluence likely has little annihilation. Each of these effects could skew estimates of the energy transfer times. For completeness, Supplementary Fig. 3 shows the resulting estimates of transfer

times recovered from all four fluences. In Supplementary Fig. 4, the same analysis as that performed on LH2-only cells is shown for LH1-only cells acquired with 17.6 μJ cm$^{-2}$, the only fluence where annihilation was observed, yielding a lifetime of 4.7 ± 0.2 ps SEM for energy transfer from LH1 to LH1.

**Model of LH1- and LH2-only membranes**. Using the recovered energy transfer times, we constructed model photosynthetic membranes. The model membranes for LH1- and LH2-only cells were constructed based on previous AFM work that showed that, in the absence of LH1s and RCs, LH2 complexes assemble into closely packed, approximately hexagonal arrangements[30]. RC-LH1-only mutants with PufX have been previously shown to form tubular membrane structures[31]. But the LH1-only mutant used in the current study lacks both RC and PufX components, so it will be composed primarily of 'empty' LH1 monomers, and it adopts a flat membrane structure, as shown in previous AFM studies[32, 33]. Initially, the model membranes are populated randomly with excitations based on the four laser fluences used. The excitations then undergo a random walk on the membrane by calculating the probability of hopping or fluorescing in a 10 fs time step. Annihilation was simulated by reducing the number of excitations to one when one complex received two or more excitations at a time. These trajectories yielded waiting time traces for the number of excited complexes that could be compared to

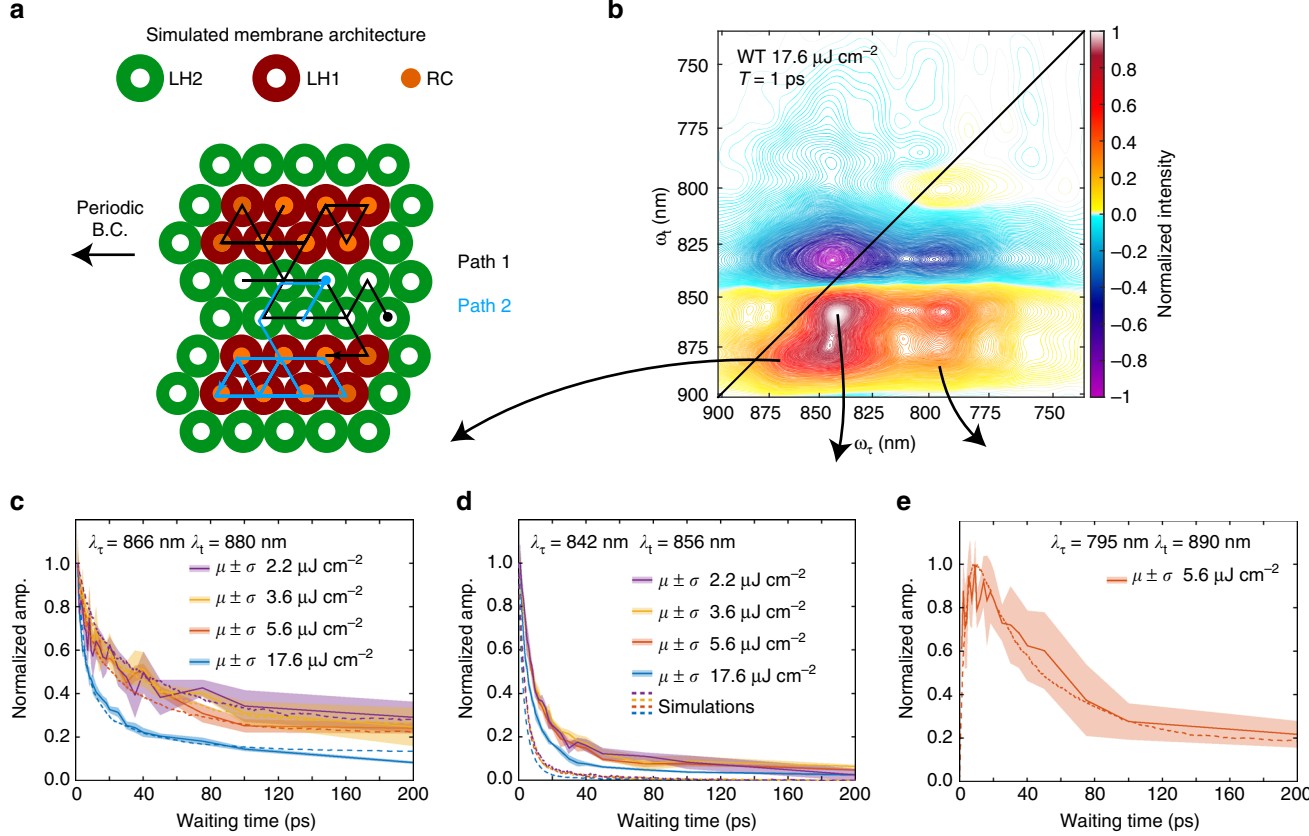

**Fig. 4** Modeling and experiment constrain membrane architecture. **a** Model of the functional light-harvesting unit and two representative simulated trajectories for the initial conditions of a single excitation in LH2 with all RCs open. **b** Absorptive 2DES spectrum of WT cells taken with 17.6 μJ cm⁻² at T = 1 ps. Waiting time traces taken from the spectral locations corresponding to **c** LH1 **d** LH2 and **e** energy transfer from LH2 to LH1 acquired at different excitation powers. The traces are the average of three scans and the shaded background is ± the standard deviation. The dashed lines show the dynamics from the model membrane. The error between model and experiment at long times in **c** is likely due to GSB from the RC special pair. The deviation at short times in **d** is likely due to spectral overlap between the LH2-LH1 cross peak and the LH2 diagonal feature

the experimental data (Fig. 1d). The domain size and fluorescence lifetimes were varied (see Supplementary Figs. 5–7). The optimal fluorescence lifetime for both LH1- and LH2-only membranes was found to be 250–300 ps. This value is in agreement with previous results[34] and also in agreement with lifetimes recovered from our LH1-only mutant data taken at low fluences that showed no annihilation and fit to a lifetime of 264 ± 7 ps SEM (Fig. 3e and Supplementary Fig. 8). While the fluorescence lifetimes of the two complexes were similar, the domain sizes were significantly different. LH2-only cells had a domain size of 64 complexes, consistent with a proposed model of LH2-only spherical vesicles[30] and LH1-only cells had a domain size of 16 complexes (Fig. 1d and Supplementary Fig. 2).

**Energy transfer dynamics in wild-type *Rba. sphaeroides*.** Using the energy transfer lifetimes (between LH2s and between LH1s) from the mutant cell studies, we can perform the same power-dependent study on wild-type (WT) cells to recover energy transfer times from LH2 to LH1 and from LH1 to the RC. The study shows annihilation only at the highest excitation fluence with annihilation appearing in both LH2 and LH1 spectral features (Fig. 4). The lack of observed annihilation dynamics at lower fluences indicates rapid transfer to LH1 and trapping via the RC. The data collected at 5.6 μJ cm⁻² is used to determine inter-complex transfer timescales to avoid complications due to annihilation. Figure 3 shows waiting time traces from the peak intensity of the GSB/SE feature of LH2 and LH1 in the WT cell data. Traces were fit to a biexponential function in the spectral

regions corresponding to LH2 and LH1, yielding energy transfer times of 4.8 ± 0.2 ps SEM from LH2 to LH1 and 49 ± 3 ps SEM from LH1 to the RC. These results are in agreement with previous measurements on membrane fragments[9, 10, 12, 21].

Combining experimental lifetimes, summarized in Supplementary Table 1, with modeling, we were able to constrain the membrane architecture and functional unit in WT *Rba. sphaeroides*. A model WT membrane was constructed in a similar manner to the LH1- and LH2-only membranes. The geometry of the model followed both electron microscopy and AFM work that has shown that the membrane forms vesicles ~ 50 nm in diameter and that RC-LH1 complexes form small domains of dimers[20, 35]. The ratio of LH1:LH2 was set to 1:1.8 to be consistent with the absorption spectrum, and the back transfer rate from LH1 to LH2 was set to ~ 120 ps to match the fluorescence spectrum in Supplementary Fig. 9. Trapping via the RC results in an oxidized special pair of bacteriochlorophyll responsible for charge separation. The oxidized special pair produces a GSB signal that overlaps the B875 GSB/SE signal. In the simulations, the signal strength of a bleached special pair was approximated to be 0.43 times the strength of the GSB/SE signal of an LH1. This approximation is based on the molar extinction coefficients for the special pair and LH1 chromophores, where the excitation was modeled as delocalized over an average of 2.5 B875 chromophores[36, 37]. The lifetime of an oxidized special pair is longer than the 200 ps trajectories simulated here, so an RC can only act as a trap for one excitation per trajectory. As in the LH1- and LH2-only models, the

membrane is initially populated with excitations based on the four excitation fluences. The excitations then undergo a random walk. During each 10 fs time step, an excitation can either transfer to another antenna complex, fluoresce, be trapped in an RC if the excitation is on an LH1 with an available RC, or annihilate with another excitation if they are on a single site, thus reducing the number of excitations on the site to one. As in the LH1- and LH2-only models, the WT model yielded waiting time dynamics that can be compared to experimental results. The domain size of LH1 complexes in an embedded array of LH2 was varied, as was the overall number of connected complexes. A functional unit comprising two domains of eight LH1 complexes embedded in 29 LH2 complexes was found to be most consistent with the 2DES data (see Fig. 4 and Supplementary Fig. 10).

## Discussion

The efficiency of the model membrane was determined by exploring trajectories where the membrane was populated with a single excitation in an LH2 complex. These trajectories represent the conditions expected during low-light growth. Figure 4a shows two of the five thousand simulated trajectories. These simulations yielded rapid transfer to the small domains of LH1. The limited back transfer rate meant that only 46% of excitations made one or more transfers from LH1 to LH2 and the majority of time before trapping or fluorescing was spent exploring the LH1/RC domains. These dynamics lead to the efficient trapping of 83% of excitations via the RC. The limited back transfer from the small pools of LH1 also restricted the excitation from exploring a large membrane environment. Given an infinite membrane, the excitation spent, on average, 61% of the time exploring one of the two nearest LH1 domains. Consequently, the functional light-harvesting unit in low-light conditions can be represented by the domain shown in Fig. 4a.

Organisms are inherently dynamic and adapt to their environment, but their responses to external conditions depend on them being alive. Photosynthetic organisms are no exception. The *Rba. sphaeroides* in this study were grown in low-light and semi-aerobic conditions and their membrane organization reflects this environment. Starved for photons, they expressed and assembled a highly efficient energy transfer pathway to funnel excitations into the RC, but this is not always the membrane organization present in *Rba. sphaeroides*. It is well established that the ratio of LH1 and LH2 changes, depending on light levels, thus altering the membrane architecture[38]. Under aerobic conditions, photo-protective mechanisms have been observed in LH1 that couple B875 to carotenoid dark states[5]. In plants and cyanobacteria, photoprotective mechanisms, such as non-photochemical quenching and state transitions, alter ultrafast energy transfer pathways and membrane architecture in response to oxidative stress, high-light conditions, and dehydration. Many questions remain on exactly how the dynamics, coupling, and architecture change under different stresses. The work presented here is a direct measure of the flow of excited states in vivo, and reveals the energy transfer pathways and membrane architecture native to *Rba. sphaeroides*. This functional model is in qualitative agreement with an atomic structural model of a low-light-adapted chromatophore vesicle from *Rba. sphaeroides*[39, 40]. Our results on bacterial cells confirm decades-old assumptions that measurements made in isolated complexes and membrane fragments resembled those of the functional unit in vivo and lay the ground work for future studies investigating ultrafast energy transfer and its regulation in living photosynthetic systems.

## Methods

**Two-dimensional electronic spectroscopy**. The excitation spectrum was produced by focusing the output of a Ti:sapphire regenerative amplifier with a 5 kHz repetition rate, 2 W output, and a 30 fs pulse duration centered at 800 nm (Coherent Inc.) into argon gas at 4 psi above atmospheric pressure. The generated supercontinuum was then spectrally and temporally shaped using an SLM-based pulse shaper (Biophotonics Solutions). The 2DES spectra were collected using the GRadient Assisted Photon Echo Spectrometer, which encodes the coherence time delay spatially across the sample. The photon echo and local oscillator fields were spectrally resolved along the rephasing axis by a spectrometer (Andor Shamrock) and heterodyne detection was achieved using a high-speed CMOS camera (Phantom Miro). Waiting times were encoded using a motorized linear stage (Aerotech).

**Data acquisition and analysis**. Raw interferograms from the camera initially have axes of coherence time and rephasing wavelength. Coherence times were sampled in 0.9 fs steps from −200 to 200 fs. Waiting time spectra were collected at 25 Hz in 0.1 fs steps for 50 fs surrounding the time points 1, 2, 3, 4, 5, 6, 7, 8, 9, 10, 12, 14, 16, 18, 20, 25, 30, 35, 40, 50, 75, 100, and 200 ps. i.e. 500 frames were taken between 0.975–1.025 ps. These 500 frames were averaged to produce a single frame representing 1 ps. By scanning the waiting time delay over 50 fs, the optical scatter, which oscillates at the optical period in the waiting time domain, is suppressed during the averaging, making the technique relatively immune to artifacts from intensely scatter light. Cubic spline interpolation was then performed to convert the single average image to axes of coherence time and rephasing frequency. An FFT then produced the image in the coherence time and rephasing time domains. The data was apodized in the rephasing time domain with a Tukey Window of width 200 fs and alpha value 0.25 centered on the photon echo signal. Next, the data was apodized in the coherence time domain with a window of width 350 fs and zero-padded to achieve approximately square frequency pixels. Lastly, the signal was shifted to zero rephasing and coherence time and a 2DFFT was performed to produce unphased 2DES spectra in the coherence frequency rephasing frequency domain. The rephasing and nonrephasing signals were then phased separately to the 1 ps pump-probe spectra following the projection slice theorem (Supplementary Fig. 11). Because pump-probe spectroscopy does not benefit from the same scatter removal techniques as 2DES, pump-probe spectroscopy was performed on isolated membrane fragments. The real valued data from the addition of the rephasing and nonrephasing spectra gave the absorptive spectra used for analysis in this manuscript. All data were triplicated in rapid succession to produce a mean valued spectrum with errors generated by the standard deviation of the three measurements. During the acquisition of the three trials, the sample was illuminated while flowing for a total of 23 min and the complete experiment time was 1.5 h per sample. All data analysis was performed using Matlab with custom software.

**Error analysis**. After data analysis, which includes filtering in Fourier domains, each point in the 2DES spectra is no longer statistically independent. The number of independent points was used to determine the standard error of the mean energy transfer and fluorescence lifetimes presented.

**Cell culture**. Mutants containing only LH1 were obtained by genomic deletion of *puc1BA* and *pufLMX*. This mutant strain lacks LH2, RC, and PufX, leading to a preferentially monomeric state of LH1. Mutants containing only LH2 were obtained by the genomic deletion of the *pufBALMX* genes. Genomic deletions were performed following the protocol described in Mothersole et al.[28]. All cells were cultured semi-aerobically in the dark at 30 °C. Prior to 2DES analysis, the cells were centrifuged (Eppendorf 5810 R 4000 rpm) and resuspended in 50/50 (v/v%) glycerol water. Analysis was performed in a 200 μm path length flow cell (Starna Cells Inc.) with a reservoir of 5 ml. Membrane fragments used only for pump-probe spectroscopy were obtained by disrupting the cells using a French press at 14,000 psi. A slow spin was performed (12,000 rpm JA 30.STI for 20 min) to remove the large cellular debris. The supernatant was diluted to an optical density of ~ 0.3 in the NIR region in the same 200 μm path length flow cell used for the 2DES analysis.

**Data availability**. The authors declare that all data supporting the findings of this study are available within the manuscript and its supplementary files or are available from the corresponding author upon request.

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

## Acknowledgements

The authors would like to thank MRSEC (DMR 14- 20709), the DARPA QuBE program (N66001-10-1-4060), AFOSR (FA9550-14-1-0367), the DoD Vannevar Bush Fellowship (N00014-16-1-2513), the Alfred P. Sloan Fellowship, the Camille and Henry Dreyfus Foundation, and the Searle Foundation for supporting the work in this publication. S.C. M. acknowledges support from the Department of Defense (DoD) through the National Defense Science & Engineering Graduate Fellowship (NDSEG) Program. P.D.D. acknowledges support from the NSF-GRFP program, and the National Institute of Biomedical Imaging And Bioengineering of the National Institutes of Health under Award Number T32-EB009412. MAA acknowledges support from the Yen Postdoctoral Fellowship and the Arnold O. Beckman Postdoctoral Fellowship. C.N.H and E.C.M. were supported by grant BB/M000265/1 from the Biotechnology and Biological Sciences Research Council (UK) and an Advanced Award from the European Research Council (338895). This research was also supported by the Photosynthetic Antenna Research Center (PARC), an Energy Frontier Research Center funded by the U.S. Department of Energy, Office of Science, Office of Basic Energy Sciences under Award Number DE-SC 0001035. That grant provided partial support for C.N.H.

## Author contributions

P.D.D. and G.S.E. designed the experiment and model, P.D.D., P.-C.T., S.C.M. and M.A.A. conducted all experiments, E.C.M. and C.N.H. developed cell strains used throughout the manuscript, P.D.D. analyzed the data, P.D.D. wrote the manuscript with participation from all authors.

## Additional information

**Competing interests:** The authors declare no competing financial interests.

