## [Peer Review File · Nature Communications]

Reviewers' comments:

Reviewer #1 (Remarks to the Author):

The authors have used 2DES to study energy flow within the LH1 and LH2 light harvesting complexes. The paper's goal is defining whether previous experiments on isolated light harvesting complexes is representative of wild type complexes. The previous experiments were prevented from studying the full wild type complexes by scattered light from the large number of different complexes. The authors overcome this limitation by using 2DES and by averaging a large number of spectra taken in very small time increments around different population times. They were then able to monitor the relaxation transients of specific 2DES features representative of LH2 and LH1 complexes. They performed these experiments on cells containing either LH1 or LH2 as well as wild type cells. The transients on cells with either LH1 or LH2 provided the hopping rates between the chromophores in each complex. The data interpretation rested heavily on the theory developed by Barzda et al (reference 26). The wild type transients provided the LH1-LH2 transfer rate. The authors then used these transfer rates to model the fluorescence lifetime of either LH1 or LH2. The authors constructed a lattice model representing complex and used random hopping and annihilation by fluorescence or trapping for different excitation fluences to correlate the measured hopping rate with the fluorescence lifetime. Their conclusions justify the previous experiments use of isolated light harvesting complexes to represent the wild type complex.

This paper is dense with information as a result of the page restrictions. Details that would help the reader understand the justification of their different choices were left out. The supporting information does help considerably but it is still a difficult paper for a reader. There are a number of examples of details that were not provided.

- The authors show that the hopping rate in LH2 is dependent on the fluence and therefore represents electronic states that are delocalized between chromophores in the complex. But then they chose the next highest fluence to determine the hopping rate instead of the lowest fluence where you would expect the hopping rate to be more representative of a photosynthetic process under solar conditions.
- There is no discussion of why they choose different kinetic models for the LH1 and LH2 only complexes and the wild type. A discussion would be appropriate for informing the reader.
- The experimental fluorescence lifetime was stated to be 264 ps but there was no reference to the origin of that value. It also seems strange that it would be the same for LH1 and LH2.
- The modeling determined that the initial reaction center population was 80% closed but the discussion of the origin of that value was not sufficient. A discussion that provided a perspective on the observation was also not adequate.

This paper is actually a very nice piece of work and an important contribution to the field. It is a topic that is interesting to the broad readership of Nature Communications. It is certainly worthy of publication, especially if the authors can address the concerns but it would benefit from not being constrained by the page limitations.

Reviewer #2 (Remarks to the Author):

Comments on

The manuscript submitted by Prof. Engel

Mapping The Ultrafast Flow Of Harvested Solar Energy In

Living Photosynthetic Cells

Peter D. Dahlberg¹, Po-Chieh Ting², Sara C. Massey², Marco A. Allodi², Elizabeth C.

Martin³, C. Neil Hunter³, and Gregory S. Engel^{2*}

The MS reports about an excellent study on ultrafast energy transfer between the antenna complexes LH1 and LH2 and the reaction center of photosynthetic cells. The authors use the latest means of technology to obtain 2D electronic absorption spectra. The 2D technique is particularly illustrative to expose spectral changes as a function of time. Very interesting is the method of the removal of scattered light. Modelling of membrane fragments is also very convincing thereby limiting the number of protein complexes involved in an energy transfer unit.

The paper may be published as is. I would just like to mention that maybe the authors could comment on the dimensions of the membrane fragments the scattering contribution of which could be taken into account.

Reviewer #3 (Remarks to the Author):

This is a very well written and interesting paper that illustrates the strength of 2D photon echo spectroscopy. By filtering the photon echo data in the Fourier domain to remove interference from light scattering, the authors have been able to measure the energy transfer in whole cells of a photosynthetic purple bacterium.

Although this is an excellent piece of work that could be published essentially as is, I do not think that the findings are of sufficiently broad significance to warrant publication in Nature. The main conclusion of the paper is that the energy transfer rates and pathways found in whole cells are the same those obtained from many years of studying isolated protein complexes. While this is an important conclusion, it does not really advance our understanding of photosynthetic energy conversion.

Experimentally, the work is an exquisite piece of spectroscopy, but 2D photon echo spectroscopy has been used to study photosynthetic energy transfer for several years now and the paper does not represent a major breakthrough in this area. This is also not the first example of an optical study of photosynthetic energy capture in whole cells. Indeed, Joliot and co-workers first reported pump-probe data of electron transfer in whole cells of cyanobacteria in the late 1990's.

Thus, I would recommend that the authors publish the paper to a more specialized biophysics journal.

Specifics:

Given the importance of the Fourier filtering to the paper it is odd that there are essentially no details of the procedure given in the manuscript or the supporting information.

Reviewers' comments:

Reviewer #1 (Remarks to the Author):

The authors have used 2DES to study energy flow within the LH1 and LH2 light harvesting complexes. The paper's goal is defining whether previous experiments on isolated light harvesting complexes is representative of wild type complexes. The previous experiments were prevented from studying the full wild type complexes by scattered light from the large number of different complexes. The authors overcome this limitation by using 2DES and by averaging a large number of spectra taken in very small time increments around different population times. They were then able to monitor the relaxation transients of specific 2DES features representative of LH2 and LH1 complexes. They performed these experiments on cells containing either LH1 or LH2 as well as wild type cells. The transients on cells with either LH1 or LH2 provided the hopping rates between the chromophores in each complex. The data interpretation rested heavily on the theory developed by Barzda et al (reference 26). The wild type transients provided the LH1-LH2 transfer rate. The authors then used these transfer rates to model the fluorescence lifetime of either LH1 or LH2. The authors constructed a lattice model representing complex and used random hopping and annihilation by fluorescence or trapping for different excitation fluences to correlate the measured hopping rate with the fluorescence lifetime. Their conclusions justify the previous experiments use of isolated light harvesting complexes to represent the wild type complex.

The authors thank the reviewer for their careful reading and consideration of our work.

This paper is dense with information as a result of the page restrictions. Details that would help the reader understand the justification of their different choices were left out. The supporting information does help considerably but it is still a difficult paper for a reader. There are a number of examples of details that were not provided.

- The authors show that the hopping rate in LH2 is dependent on the fluence and therefore represents electronic states that are delocalized between chromophores in the complex. But then they chose the next highest fluence to determine the hopping rate instead of the lowest fluence where you would expect the hopping rate to be more representative of a photosynthetic process under solar conditions.

There is a subtle distinction between the two different rates mention in the manuscript: annihilation rate and hopping rate. The hopping rate is independent of the excitation fluence; the annihilation rate however is highly dependent on the excitation fluence and is used to recover the hopping rate. The hopping rate could be obtained from equations 1 and 2 from data collected at just one fluence and indeed in the manuscript uses the data obtained from the fluence of 5.6

$\mu\text{J}/\text{cm}^2$ in the case of LH2 as the reviewer mentioned. The reason to do multiple excitation fluences is to confirm the presence of annihilation to determine the annihilation rate at a fluence with appreciable annihilation.

The data collected at $5.6 \mu\text{J}/\text{cm}^2$ was used to calculate the hopping rate in the manuscript to avoid multiple initial excitations within a single LH2 that would have occurred, albeit rarely, in the highest fluence data. At the lowest fluence there is quite little annihilation making it difficult to use the annihilation rate to recover the hopping rate. This discussion, previously absent from the manuscript, has been amended with the following sections, as well as an additional SI figure that shows the recovered lifetimes from the 4 different fluences used.

“Performing a power-dependence study of dynamics with excitation fluences of 17.6, 5.6, 3.6, and $2.2 \mu\text{J}/\text{cm}^2$ per beam allows us to identify annihilation processes by observing changes in dynamics between excitation fluences and then to exploit the annihilation dynamics to determine energy transfer times between the isoenergetic complexes.”

“The recovered lifetime of 2.7 ± 0.1 ps is homogenous and the same for the ESA feature above the diagonal as well as the GSB/SE feature below the diagonal. Data collected at any of these excitation fluences could have been used to estimate the energy transfer time between LH2s, however the highest fluence used risks initially populating LH2s with multiple excitations that would give rise to annihilation not due to inter-complex transfer and the lowest fluence likely has little annihilation. Each of these effects could skew estimates of the energy transfer times. For completeness, Supplementary Fig. 3 shows the resulting estimates of transfer times recovered from all 4 fluences.”

Figure S3: Histogram of the energy transfer times between LH2s recovered from 2DES data of LH2 only cells at fluences of 17.6, 5.6, 3.6, and 2.6 $\mu\text{J}/\text{cm}^2$.

•There is no discussion of why they choose different kinetic models for the LH1 and LH2 only complexes and the wild type. A discussion would be appropriate for informing the reader.

The geometry of the models is based on previous AFM work. The following discussion of this prior work has been added to make the manuscript more broadly readable and informative. Additionally, the initial submission of the manuscript had the LH1-only membrane arranged in dimers. The mutant used in this study lacks the protein PufX responsible for dimerization. The simulations for LH1-only cells have been recalculated for more square arrangement of complexes rather than a string of dimers, though the results change only slightly. Supplementary figures S2, S5, and S7 have been altered to reflect these changes.

“The model membranes for LH1- and LH2-only cells were constructed based on previous AFM work that showed LH2 complexes in the absence of LH1s and RCs assemble with approximately hexagonal close packing.³⁰ LH1-only mutants with PufX have been shown

to form tubular membrane structures,³¹ but the LH1-only mutant used in this study lacks PufX, meaning that it will be composed primarily of monomers and will likely adopt a flat membrane structure.”

•The experimental fluorescence lifetime was stated to be 264 ps but there was no reference to the origin of that value. It also seems strange that it would be the same for LH1 and LH2.

The fluorescence lifetime of 264 ps was determined from exponential fits to the low power experiments conducted in LH1 only cells that showed no annihilation. The results of these fits can be seen in Figure 3e of the main text as well as supplementary figure 8. Because annihilation in the LH2-only cells was present at all fluences we used 264 ps to approximate the lifetime of LH2. This lifetime for LH2 was in agreement with literature values, a reference to this work was lost during the revision process. We thank the reviewer for their close reading of the text and have added the reference back in.

Hunter, C. N., Bergstrom, H., van Grondelle, R. & Sundstrom, V. Energy-transfer dynamics in three light-harvesting mutants of *Rhodobacter sphaeroides*: a picosecond spectroscopy study. *Biochemistry* **29**, 3203-3207, doi:Doi 10.1021/Bi00465a008 (1990).

From the manuscript cited above the lifetime of LH2 at 850 nm is 273 ps. In order to reduce future confusion we have redone the modeling of the wild type cells using a fluorescence lifetime of 264 ps for LH1 and 273 ps for LH2. This had no measurable effect on the results. We have added the following text in the caption to table S1

Complex-Complex	τ_{hop} (ps)
LH2→LH2	2.7 ± 0.1
LH2→LH1	4.8 ± 0.2
LH1→LH1	4.7 ± 0.2
LH1→RC	49 ± 3
LH2 fluorescence	273^1
LH1 fluorescence	264 ± 7

Supplementary Table 1: Energy transfer times between complexes recovered from annihilation studies in mutant *Rba. sphaeroides* and biexponential fits to low fluence scans. Because annihilation was present at all powers in the LH2 only cells, the fluorescence lifetime of LH2 from Hunter et al.¹ was used for modeling.

- 1 Hunter, C. N., Bergstrom, H., van Grondelle, R. & Sundstrom, V. Energy-transfer dynamics in three light-harvesting mutants of *Rhodobacter sphaeroides*: a picosecond spectroscopy study. *Biochemistry* **29**, 3203-3207, doi:Doi 10.1021/Bi00465a008 (1990).

•The modeling determined that the initial reaction center population was 80% closed but the discussion of the origin of that value was not sufficient. A discussion that provided a perspective on the observation was also not adequate.

We would like to thank the reviewer for raising this issue. It motivated us to take a closer look at the assumptions we had made surrounding the closing of the reaction centers. Initially we had assumed that the GSB signal of the special pair in the RC would be negligible in comparison to the signals from the LH1 and LH2, however, upon closer inspection we find that incorporating this effect into our simulations yields results more consistent with the experimental data and has none of the RCs in the closed state. This alteration to the simulation does not change which membrane geometry agrees best with experimental data, but it does change the simulated trajectories. We have added the following discussion to the text

“Combining experimental lifetimes, summarized in Supplementary Table 1, with modeling, we were able to constrain the membrane architecture and functional unit in WT *Rba. sphaeroides*. A model WT membrane was constructed in a similar manner to the LH1- and LH2-only membranes. The geometry of the model followed both EM and AFM work that has shown that the membrane forms vesicles ~ 50 nm in diameter and that LH1 exists in small domains of dimers.^{20,33} The ratio of LH1:LH2 was set to 1:1.8 to be consistent with the absorption spectrum, and the back transfer rate from LH1 to LH2 was set to ~120 ps to match the fluorescence spectrum in Supplementary Fig. 9. Trapping via the RC results in an oxidized special pair of bacteriochlorophyll responsible for charge separation. The oxidized special pair produces a GSB signal that overlaps the B875 GSB/SE signal. In the simulations, the signal strength of a bleached special pair was approximated to be 0.43 times the strength of GSB/SE of an LH1. This approximation is based on the molar extinction coefficients for the special pair and LH1 chromophores where the excitation was modeled as delocalized over an average of 2.5 B875 chromophores.^{34,35} The lifetime of an oxidized special pair is longer than the 200 ps trajectories simulated here, so an RC can only act as a trap for 1 excitation per trajectory. As in the LH1- and LH2-only models, the membrane is initially populated with excitations based on the four excitation fluences. The excitations then undergo a random walk. During each 10 fs time step an excitation can either transfer to another antenna complex, fluoresce, be trapped in an RC if the excitation is on an LH1 with an available RC, or annihilate with another excitation if they are on a single site thus reducing the number of excitations on the site to one. As in the LH1- and LH2-only models the wild type model yielded waiting time dynamics that can be compared to experimental results. The domain size of LH1 complexes in an embedded array of LH2 was varied, as was the

overall number of connected complexes. A functional unit of 2 domains of 8 LH1 complexes embedded in 29 LH2 complexes was found to be most consistent with the 2DES data, see Fig. 4 and Supplementary Fig. 10.”

We have also revised figure 4 and S10 to reflect the changes in the simulations.

This paper is actually a very nice piece of work and an important contribution to the field. It is a topic that is interesting to the broad readership of Nature Communications. It is certainly worthy of publication, especially if the authors can address the concerns but it would benefit from not being constrained by the page limitations.

We thank the reviewer again for their careful consideration of our work and essential feedback. We hope the discussions added to the text as well as additional SI figure make the manuscript more accessible to the broad readership of Nature Communications.

Reviewer #2 (Remarks to the Author):

Comments on

The manuscript submitted by Prof. Engel

Mapping The Ultrafast Flow Of Harvested Solar Energy In
Living Photosynthetic Cells

Peter D. Dahlberg¹, Po-Chieh Ting², Sara C. Massey², Marco A. Allodi², Elizabeth C.
Martin³, C. Neil Hunter³, and Gregory S. Engel^{2*}

The MS reports about an excellent study on ultrafast energy transfer between the antenna complexes LH1 and LH2 and the reaction center of photosynthetic cells. The authors use the latest means of technology to obtain 2D electronic absorption spectra. The 2D technique is particularly illustrative to expose spectral changes as a function of time. Very interesting is the method of the removal of scattered light. Modelling of membrane fragments is also very convincing thereby limiting the number of protein complexes involved in an energy transfer unit. The paper may be published as is. I would just like to mention that maybe the authors could comment on the dimensions of the membrane fragments the scattering contribution of which could be taken into account.

The authors thank the reviewer for their close reading of the text.

Reviewer #3 (Remarks to the Author):

This is a very well written and interesting paper that illustrates the strength of 2D photon echo spectroscopy. By filtering the photon echo data in the Fourier domain to remove interference

from light scattering, the authors have been able to measure the energy transfer in whole cells of a photosynthetic purple bacterium.

The authors thank the reviewer for their time and consideration of our work.

Although this is an excellent piece of work that could be published essentially as is, I do not think that the findings are of sufficiently broad significance to warrant publication in Nature. The main conclusion of the paper is that the energy transfer rates and pathways found in whole cells are the same those obtained from many years of studying isolated protein complexes. While this is an important conclusion, it does not really advance our understanding of photosynthetic energy conversion.

Experimentally, the work is an exquisite piece of spectroscopy, but 2D photon echo spectroscopy has been used to study photosynthetic energy transfer for several years now and the paper does not represent a major breakthrough in this area. This is also not the first example of an optical study of photosynthetic energy capture in whole cells. Indeed, Joliot and co-workers first reported pump-probe data of electron transfer in whole cells of cyanobacteria in the late 1990's. Thus, I would recommend that the authors publish the paper to a more specialized biophysics journal.

Specifics:

Given the importance of the Fourier filtering to the paper it is odd that there are essentially no details of the procedure given in the manuscript or the supporting information.

The authors agree and have added significantly more details to the methods section to make the work more easily reproduced and evaluated.

“Data acquisition and analysis: Raw interferograms from the camera initially have axes of coherence time and rephasing wavelength. Coherence times were sampled in 0.9 fs steps from -200 to 200 fs. Waiting spectra were collected at 25 Hz in 0.1 fs steps for 50 fs surrounding the time points 1, 2, 3, 4, 5, 6, 7, 8, 9, 10, 12, 14, 16, 18, 20, 25, 30, 35, 40, 50, 75, 100, and 200 ps. i.e. 500 frames were taken between 0.975-1.025 ps. These 500 frames were averaged to produce a single frame representing 1 ps. By scanning the waiting time delay over 50 fs, the optical scatter, which oscillates at the optical period in the waiting time domain, is suppressed during the averaging, making the technique relatively immune to artifacts from intensely scatter light. Cubic spline interpolation was then performed to convert the single average image to axes of coherence time and rephasing frequency. An FFT then produced the image in the coherence time and rephasing time domains. The data was apodized in the rephasing time domain with a Tukey Window of width 200 fs and alpha value 0.25 centered on the

photon echo signal. Next, the data was apodized in the coherence time domain with a window of width 350 fs and zero-padded to achieve approximately square frequency pixels. Lastly, the signal was shifted to zero rephasing and coherence time and a 2DFFT was performed to produce unphased 2DES spectra in the coherence frequency rephasing frequency domain. The rephasing and nonrephasing signals were then phased separately to the 1 ps pump-probe spectra following the projection slice theorem (Supplementary Fig. 11). Because pump-probe spectroscopy does not benefit from the same scatter removal techniques as 2DES, pump-probe spectroscopy was performed on isolated membrane fragments. The real valued data from the addition of the rephasing and nonrephasing spectra gave the absorptive spectra used for analysis in this manuscript. All data were triplicated in rapid succession to produce a mean valued spectrum with errors generated by the standard deviation of the three measurements. During the acquisition of the three trials, the sample was illuminated while flowing for a total of 23 minutes the complete experiment time was 1.5 hours per sample. All data analysis was performed using Matlab with custom software.”

REVIEWERS' COMMENTS:

Reviewer #1 (Remarks to the Author):

The author has addressed the issues that I was concerned about and the manuscript is suitable for publication in Nature Communications. It is an excellent piece of work and it is a topic of broad interest, certainly suitable for Nature Communications.

Reviewer #1 (Remarks to the Author):

The author has addressed the issues that I was concerned about and the manuscript is suitable for publication in Nature Communications. It is an excellent piece of work and it is a topic of broad interest, certainly suitable for Nature Communications.

We thank the reviewer for their careful reading of the text and kind remarks.